# Potassium Application Boosts Photosynthesis and Sorbitol Biosynthesis and Accelerates Cold Acclimation of Common Plantain (*Plantago major* L.)

**DOI:** 10.3390/plants9101259

**Published:** 2020-09-24

**Authors:** Li-Hsuan Ho, Regina Rode, Maike Siegel, Frank Reinhardt, H. Ekkehard Neuhaus, Jean-Claude Yvin, Sylvain Pluchon, Seyed Abdollah Hosseini, Benjamin Pommerrenig

**Affiliations:** 1Plant Physiology, University Kaiserslautern, Paul-Ehrlich-Str., 67663 Kaiserlautern, Germany; lihsuan@rhrk.uni-kl.de (L.-H.H.); rode@rhrk.uni-kl.de (R.R.); maimue@rhrk.uni-kl.de (M.S.); frank.reinhardt@rr-home.de (F.R.); neuhaus@rhrk.uni-kl.de (H.E.N.); 2Centre Mondial de l’Innovation Roullier—Laboratoire de Nutrition Végétale, 18 avenue Franklin Roosevelt 35400 Saint-Malo, France; JeanClaude.Yvin@roullier.com (J.-C.Y.); Sylvain.Pluchon@roullier.com (S.P.); SeyedAbdollah.Hosseini@roullier.com (S.A.H.)

**Keywords:** potassium, sucrose, sorbitol, phloem, Plantago, photosynthesis, cold stress

## Abstract

Potassium (K) is essential for the processes critical for plant performance, including photosynthesis, carbon assimilation, and response to stress. K also influences translocation of sugars in the phloem and regulates sucrose metabolism. Several plant species synthesize polyols and transport these sugar alcohols from source to sink tissues. Limited knowledge exists about the involvement of K in the above processes in polyol-translocating plants. We, therefore, studied K effects in *Plantago major*, a species that accumulates the polyol sorbitol to high concentrations. We grew *P. major* plants on soil substrate adjusted to low-, medium-, or high-potassium conditions. We found that biomass, seed yield, and leaf tissue K contents increased in a soil K-dependent manner. K gradually increased the photosynthetic efficiency and decreased the non-photochemical quenching. Concomitantly, sorbitol levels and sorbitol to sucrose ratio in leaves and phloem sap increased in a K-dependent manner. K supply also fostered plant cold acclimation. High soil K levels mitigated loss of water from leaves in the cold and supported cold-dependent sugar and sorbitol accumulation. We hypothesize that with increased K nutrition, *P. major* preferentially channels photosynthesis-derived electrons into sorbitol biosynthesis and that this increased sorbitol is supportive for sink development and as a protective solute, during abiotic stress.

## 1. Introduction

Potassium belongs to the group of major plant nutrients and accumulates to high levels in nearly all plant tissues [1]. In most types of soils, potassium is relatively highly abundant, although only about 2% of total soil potassium is available to the plant [2]. This is because most (up to 98%) potassium is tightly bound to minerals like feldspar, feldspathoid (including nepheline and leucite), mica, or clay [2]. While clay-fixed potassium might serve as a reservoir for K when plants grow under K-limiting conditions, only the cationic potassium (K^+^ form, representing between 1–2 % of total K) is directly available for uptake into roots. In contrast to other highly abundant nutrients like phosphate, sulfate, or nitrogen, K does not become part of a certain cellular structure or molecule. Nevertheless, K serves as an essential ion required for a wide number of different processes that are critical for plant development and performance. K has crucial functions in enzyme activation, osmotic adjustment, regulation of turgor pressure, and membrane electric potential [3]. Most prominent is the important and quite well-characterized function of K for the control of stomatal function, where potassium, besides malic acid, is a major solute that accumulates in guard cell vacuoles, during stomatal opening [4,5]. Accordingly, plants grown under limiting potassium supply have difficulties in controlling their “water use efficiency”. Given the overall impact of a balanced water-use efficiency and the interaction between plant K levels and water loss, it is understandable that impaired potassium homeostasis additionally plays a major role in the long distance transport of water and other nutrients within the xylem system [6,7]. In particular, reduced translocation of water itself as well as of nitrate, phosphates, calcium, and magnesium, is caused by K limitation and leads to further pleiotropic plant phenotypes, most notably, reduced productivity, in case of crops species [8,9].

Potassium represents the major cation not only in the xylem but also in the phloem [1,10,11]. The phloem functions as the long-distance transport system for fixed carbon in plants connecting source (leaves) and sink (root and storage) organs. A balanced long-distance transport of sucrose from source to sink tissues in the phloem, comprising the sieve elements and companion cells, is highly critical for a correct plant development and high productivity [12,13]. Potassium plays a crucial role in the efficient translocation of sucrose in the phloem, and K-deficiency results in diminished phloem sugar transport from source to sink tissues [14,15]. There are two explanations for this. First, K ensures efficient water flow in the xylem by establishing positive root pressure. Xylem-derived water influx into phloem elements in shoots, then drives mass flow of sugars and assimilates back to sink organs. Second, K release from the phloem to the apoplast acts as an auxiliary power supply for sugar translocation, when cellular energy in the form of ATP is low, e.g., in stem or root tissue where ATP generation is limited by the low amount of photosynthetically active mesophyll cells in this tissues [16,17]. K supply might therefore become increasingly important for sugar translocation in plants with long internodal transport sections, or under conditions where drastic sugar accumulation plays a pivotal role for plant fitness and survival, most importantly under drought or cold conditions [18,19].

Cold stress is a major abiotic stress for crop plants causing different symptoms like reduced leaf expansion, wilting, and chlorosis [20]. Cold stress also induces reactive oxygen species (ROS) formation because excess photosynthesis-derived electrons cannot be incorporated into reduced sugars, because of the slowdown of Calvin-Benson cycle enzymes under low temperatures [19,21]. ROS can severely damage thylakoids and other intracellular membranes like the tonoplast, causing bursting of vacuoles and flooding of the cytosol with protons and digestive enzymes [19]. Sugars, but also other oligomeric carbohydrates from the raffinose oligosaccharide or fructan family can alleviate cold stress symptoms by scavenging ROS [19,22,23].

Many plant species, including but not limited to the *Rosaceae* family, accumulate polyols or sugar alcohols, which represent highly reduced forms of glucose or fructose. In addition to sucrose, they are also transported in the phloem of celery (*Apium graveolens* L.), peach (*Prunus persica* (L.) Batsch), or different plantain species (*Plantaginaceae*). Polyols are rich in stored reducing power and, like the above-mentioned compounds, potent ROS quenchers [24]. They are, therefore, not only attractive building blocks for plants providing both energy and carbon to sink organs but also compounds formed to mitigate abiotic and biotic stresses [25,26]. Common plantain (*Plantago major* L.) is a robust, fast-growing perennial weedy plant, which accumulates polyol sorbitol to more than ten times higher concentrations, in comparison to sucrose in the leaves [26,27]. The relative amounts of sucrose and sorbitol loaded into the *P. major* phloem companion cells could be adjusted by the expression and activity of the corresponding sucrose and sorbitol phloem loading transporters [26,28] and are altered under stress conditions like salt stress [26] or low soil boron conditions [11].

K plays a very important role in the mitigation of different abiotic stresses, such as drought, heat, and cold stress [29,30,31]. In this context, the role of K in mitigating the adverse effect of cold stress was reported in ginseng (*Panax ginseng* Russ) by increasing the antioxidant levels and reducing the production of ROS [32]. It was also shown that sufficient K supply effectively protects plants against freezing temperatures [30]. Given the positive effects of both K and sugars on abiotic stress, we asked whether K-dependent stress mitigation could function via enhanced sugar or sugar alcohol accumulation. In this study, an attempt was undertaken to investigate the effects of different soil K supplies on growth and sugar metabolism of *P. major*, as a model plant containing high amounts of the polyol sorbitol. We aimed to explore how potentially K-dependent alterations of sugar and sorbitol levels would affect *P. major* cold stress response. We found that the higher application of K induced sugar metabolism in particular accumulation of sorbitol and accelerated cold acclimation.

## 2. Results

### 2.1. K Promotes Biomass Formation in Both Source and Sink Organs

For analysis of K-dependent effects, *P*. *major* plants were grown on a soil substrate with extremely low nutritional value used in previous studies [11,33]. This allowed for the adjustment of the required nutrients for *P*. *major* growth. The substrate was supplemented with three different doses of soil K: K1 as low K (= 27 mg K/kg soil substrate), K2 as optimum K (= 142 mg K/kg soil substrate), and K3 as high K (= 500 mg K/kg soil substrate). The differences in the soil K levels resulted in differential growth behavior of the *P*. *major* plants (Figure 1). Plants supplemented with optimum or high K were larger (Figure 1A) and developed new leaves faster than the plants grown on low K (Figure 1B). Both shoot fresh and dry weights of plants grown on K2 soil were almost twice as high, compared to K1 plants, and the increase was even more pronounced in K3 plants (Figure 1C,D). Water content of shoots also increased significantly with increasing K supply in the soil substrate (Figure 1E). Higher K also promoted growth of inflorescences, seed yield per inflorescence, and root biomass, in a dose-dependent manner (Figure 1F–H). These data suggested that the K-dependent increase in seed yield and root biomass resulted from increased provision of these sink organs, with carbohydrates as building blocks.

### 2.2. K Alters the Inorganic Ion Levels of Leaves

The different K supply resulted in the regulation of different nutrients in shoots (Figure 2). The shoot K concentration increased differentially in soil, in a K-dependent manner. This trend was expected due to different levels of K in the substrate. K2 plants accumulated about 20% more K than K1 plants, while K3 plants accumulated about 17% more K than K2 plants, indicating the higher capacity of the plants to accumulate more K when the K level increased in the soil substrate (Figure 2A). Unlike K, the levels of Na were 30- to 80-times lower in shoots, irrespective of the K level in the soil substrate (Figure 2B). K1 plants accumulated twice as much Na than K2 and K3 plants, reflecting the supplementation of K_2_HPO_4_ with Na_2_HPO_4_ in the K1 substrate and suggesting the replacement of K with Na, under the low-K condition (Figure 1B). Like Na, both divalent cation magnesium (Mg^2+^) and calcium (Ca^2+^), decreased with higher K soil and tissue concentration, which was expected under the low K condition (Figure 2C,D). Such opposite accumulation of Mg^2+^, Ca^2+^, and K^+^ under high K application was already reported, for e.g., in maize, poplar, and green bean [34,35,36] and was described as “cation antagonism” [37,38]. Significant differences were also detected for the two nitrogen forms—ammonium (NH_4_^+^) and nitrate (NO_3_^−^) (Figure 2E,F). The concentration of both nutrients reached higher values in K2 plants than in K1 and K3 plants. The phosphate (PO_4_^3−^) levels were marginally but significantly higher in K3, in comparison to K2 plants (Figure 2G). The sulfate (SO_4_^2−^) contents of K3 plants were lower than those of K1 and K2 plants (Figure 2H). Compared to K1 plants, the chloride levels were significantly increased in K3, but not in K2 plants (Figure 2I). The high Cl^−^ contents of K3 plants were most likely due to the application of KCl as a form of K. The Cl^−^ levels were identical in both K1 and K2 grown plants, suggesting that the observed differences in growth behavior and ion accumulation between K1 and K2 were not caused by a differential Cl^−^ accumulation of these plants.

### 2.3. High K Levels Boost Photosynthesis and Reduce Non Photochemical Quenching

We measured photosystem activity in dependence of the K supply level, with pulse amplitude modulated (PAM) fluorometry. The K nutritional status of the plants influenced electron transfer rates (ETR), photosystem II activities [Y(II)], non-photochemical quenching [Y(NPQ)], and non-regulated energy dissipation [Y(NO)] (Figure 3). ETR increased with light intensity in a K-dependent manner (Figure 3A). The increased ETR induced a slight increase in Y(II) in K2 plants, compared to K1 plants (Figure 3B). The higher ETR of K3 plants resulted in higher PAR-dependent Y(II) percentages, in comparison to K2 and K1 plants (Figure 3B). This indicates that higher than adequate K was required for the boosting of the photosynthesis machinery. The high K supply of K3 plants also resulted in markedly reduced Y(NPQ) values, in comparison to K2 and K1 plants, suggesting lower excess electrons in photosystems, especially under high-light conditions (Figure 3C). K3 plants had a higher percentage of Y(NO) representing the amount of unused energy (reflected photons) in comparison to K2 or K1 plants (Figure 3D). The levels of both chlorophyll A or B did not significantly differ between the different K levels in the soil substrate (Figure 3E). As NPQ can in part be dependent on the epoxidation state of photoprotective xanthophylls, we determined the combined levels of the epoxidized violaxanthin (VX), the partly de-epoxidized antheraxanthin (AX), and the fully de-epoxidized zeaxanthin (Zx) (Figure 3F), and the time-dependent formation of Zx, after the transfer of dark-adapted leaves to high light (750 µE) (Figure 3G). The measurements revealed that both xanthophyll levels and the kinetics of the synthesis of photoprotective Zx, were not significantly different between the three K conditions (Figure 3F,G).

### 2.4. K Effects on Leaf Carbon Dioxide and Water Exchange

The observed increase in photosynthesis and plant growth under higher soil K conditions encouraged us to record the gas exchange parameters and analyze soluble sugars, sorbitol, as well as the starch concentrations in leaves. Carbon assimilation rate in light (*A*) increased with increasing K level in the soil substrate and was twice as high in the K3 plants, in comparison to the K1 plants (Figure 4A). Stomatal conductance (g) in the light was not significantly influenced by the different K treatments (Figure 4B). The higher soil K level promoted sorbitol accumulation in leaves (Figure 4C). While the sucrose levels were not significantly different between the K treatments (Figure 4D), glucose levels decreased in the K2 and K3 plants, compared to the K1 plants (Figure 4E). In addition, starch levels also decreased in a K-dependent manner (Figure 4F). These data showed that higher K application led to a higher accumulation of sorbitol and supported a higher carbon assimilation and, thus, boosted plant growth.

### 2.5. Phloem Exudate Composition Under Different K Supply

To analyze the effect of K supply on phloem sap sugar composition, phloem exudates were collected from the K1, K2, and K3 plants. (Figure 5). The amounts of K exuded from the phloem only slightly increased in a K-dependent manner (Figure 5A), but did not reflect the greater differences observed for whole leaves (Figure 2A). The slight increase in phloem K was accompanied by a gradual decrease of phloem Ca^2+^ (Figure 5B). The sugar profile of the phloem sap differed from that of whole leaves. In this context, the total level of sucrose and sorbitol as non-reducing sugars, which are the main sugars in phloem of *P*. *major* [26,27] did not differ between the three soil K conditions (Figure 5C). However, the ratio of sorbitol to sucrose released from the phloem, increased with higher K in the soil (Figure 5D). Moreover, monosaccharides glucose and fructose, which are not transported in the phloem, were present in lower amounts than sucrose or sorbitol (Figure 5E).

### 2.6. Cold Acclimation Kinetics under Different Soil K Levels

Sugars and sugar alcohols accumulate in many plant species to high amounts during stress conditions [19,39] and fulfil important protective functions, especially during drought or low temperatures [21,40,41]. As K influenced sugar and sorbitol accumulation in *P. major*, we asked whether and to what extent the different K nutritional status would influence cold acclimation of *P. major* plants. To analyze such effects on the accumulation kinetic of sugars, plants grown on the different K soil levels were transferred from standard 20 °C conditions to 4 °C and monitored for their water and sugar levels (Figure 6). Before transferring the plants to cold (day 0), both K2 and K3 plants had similar water contents, which were slightly above those of K1 plants. Moving the plants to low temperatures resulted in the wilting of leaves. (Figure 6A). Notably, the degree of wilting imposed by cold treatment was dependent on the soil K level. Indeed, the K1 plants lost almost 40% of their tissue water during the first two days in the cold and did not fully recover over a period of five days. The water loss was less severe in the K2 and K3 plants, with K2 plants losing about 25% and K3 plants about 15% water, during the first two days. The water content of both K2 and K3 plants was restored to pre-stress levels on the fifth day in the cold (Figure 6B). Before transferring the plants to cold condition, the sorbitol concentrations in K3 plants were at least 50% higher than those of K1 and K2 plants. After transferring to cold condition, plants started to accumulate sorbitol, as well as monosaccharides glucose and fructose (Figure 6C–F). On the following days, sorbitol contents of K2 and K3 plants remained above those of K1, with K3 plants showing the highest contents, and this difference coincided with the shoot water content of the plants (Figure 6C). After five days in the cold, glucose accumulated to about 180 µmol and fructose to about 130 µmol per g DW. The accumulation of both monosaccharides occurred faster in K3 plants, when compared to K1 and K2. The K3 plants already reached high levels on the second day after cold exposure, exhibiting almost twice as high concentrations of glucose and fructose, when compared to the K2 and K1 plants (Figure 6D,E). Sucrose accumulated to lower amounts under low temperature, but with similar kinetics in both K1 and K2 plants (Figure 6F). In K3 plants, sucrose concentration peaked at day one, after transfer to the cold condition and decreased again at day two. This was coincident with the higher accumulation of monosaccharides at the same time-point, suggesting hydrolysis of sucrose into glucose and fructose monomers. It is worthy to note that the maltose levels sharply increased more than twentyfold, after the transfer of the plants to cold condition, irrespective of the K level in the soil substrate (Figure 6G). In contrast, the starch levels decreased in a cold-exposure-dependent manner and reached the lowest levels at day five, after transferring to the cold condition (Figure 6H). Together, these results indicate that the luxury K supply in K3 plants increased the accumulation rate of sugars in plants, in response to the cold stress.

### 2.7. Expression of Sorbitol and Sucrose Transporter Genes under Different Levels of Soil K

In the next step, we determined the expression of the genes involved in sorbitol, loading into the phloem (*PmPMT1* and *PmPMT2*) as well as *PmSUC2* as the phloem sucrose loader. In addition, the expression levels of sorbitol dehydrogenase (SDH) that regulates sorbitol accumulation was measured (Figure 7). Both soil K level and environmental temperature influenced the expression of all tested genes to some extent. *SDH* expression levels remained unchanged at 20 °C but were markedly reduced at 4 °C (Figure 7A), independent of the soil K level. The expression of sucrose transporter *PmSUC2* and sorbitol transporters *PmPMT1* and *PmPMT2* were constant at 20 °C, irrespective of the level of K in the soil (Figure 7B–D). Air temperature did not significantly alter the *SUC2* expression levels, but increased the expression of both sorbitol transporter genes (Figure 7B). Interestingly, this increase was dependent on the K soil level and higher K levels dampened the induction in the cold. (Figure 7C,D). These data showed that the K level changed the expression level sorbitol-related genes, in a temperature-dependent manner.

## 3. Discussion

The role of mineral nutrition in carbohydrate metabolism and partitioning is well-established [42]. A significant number of studies especially demonstrated the importance of K in carbohydrate metabolism and transport [7,14,15,43,44]. A regulated carbohydrate and most importantly sugar accumulation are crucial for acclimation and adaption of plants to changing environmental conditions [19], and therefore, links K nutrition and the ability of plants to cope with biotic and abiotic stresses. This is especially relevant during cold stress, where sugars can serve a dual protective function, as both anti-freeze agents and ROS scavengers [19,40,45]. In this study an attempt was undertaken to investigate the dose-dependent role of K in *P. major* as a high accumulator of sorbitol under cold stress. We showed that *P. major* responded clearly in a dose-dependent manner to K nutrition and coped against low temperature, by increasing soluble sugar and sorbitol levels.

Numerous studies indicate that the application of K boosts plant growth and development and avoids yield penalty. It was shown that application of higher K doses improved growth and increased the biomass of the source and sink organs of different plant species like tomato (*Solanum lycopersicum* L.), sweet potato (*Ipomoea batatas* L.), and barley (*Hordeum vulgare* L.) [31,46,47]. In line with these reports, we found that *P. major* also responded positively to application of K, where growth and development, as well as seed yield, increased with higher doses of K (Figure 1). In the present work, the K-dependent increase of biomass correlated with photosynthesis and carbon fixation, which both increased in a K-dependent manner (Figure 3 and Figure 4). ETR, Y(II), and NPQ values clearly indicated that photosynthetic efficiency benefited from additional K fertilization (Figure 3). The lower NPQ in plants grown under the luxury K (K3) supply indicated that these plants were most probably less exposed to ROS, in comparison to plants grown under low (K1) and intermediate K (K2) supplies. Plants might benefit from this K-dependent reduction of ROS pressure during the applied cold stress treatment (Figure 6). NPQ can be caused by the diversion of excess electrons to xanthophylls and carotenoids [48,49]. The xanthophyll cycle shuttles electrons between the epoxidized violaxanthin (VX), the partly de-epoxidized antheraxanthin (AX), and the fully de-epoxidized zeaxanthin (Zx). De-epoxidized xanthophylls delocalize electrons in their aromatic ring systems and contribute to the protection of photodamage of the photosynthetic machinery. Neither levels or formation kinetics of Zx in the light were significantly different among the K treatments (Figure 3) indicating that plants that did receive higher K had the same xanthophyll-dependent quenching capacity, as plants grown with lower K supply. The measured Y(NPQ) was in part associated with the dissipation of photosynthetic energy (mostly as heat) in the Mehler–Ascorbate pathway (water cycle), in which excess electrons were directly diverted from PSI to water molecules, without using them for the formation of reduction equivalents, and therefore, reduced sugars [50]. The reduced overall NPQ, unchanged xanthophylls, and increased sorbitol accumulation in high K-grown plants suggested that the excess photons were diverted to a lesser extent to the water cycle, under this condition. Sorbitol biosynthesis requires reduction of glucose-6-phosphate and is, therefore, a sink for electrons transferred by NADPH/H^+^ [26]_._ The increased sorbitol biosynthesis in plants grown with high K (Figure 4 and Figure 6) suggested that sorbitol biosynthesis functioned as an acceptor for excess photosynthesis-derived electrons, especially during cold conditions (Figure 6). The ability to synthesize sorbitol, a maximally reduced sugar, probably enables *P. major* to also efficiently recycle NADPH to NADP^+^ under low temperature-induced electron excess.

Sorbitol itself is a powerful osmolyte and is preferentially loaded into Plantago phloem over sucrose, under osmotically challenging conditions [26]. The higher content of sorbitol in the leaves of *P. major* plants grown under high K in this study might also be the reason for the higher water content in these plants, under normal conditions (Figure 1). In the present work, glucose and starch accumulated under low K supply in leaves, indicated that the movement of sugar to sink organs was blocked under such condition. In contrast, the level of sorbitol increased under high supply of K, in a dose-dependent manner. The observed K-dependent shift towards a higher sorbitol to sucrose ratio, supported a scenario where higher photosynthetic rates led to increased amount of reducing power, in the form of NADPH necessary for a complete reduction of glucose via sorbitol biosynthetic pathways. A similar observation was made in pear, a member of the sorbitol accumulating and translocating Rosaceae family, where K application had positive effects on sorbitol accumulation and induced sorbitol biosynthesis genes [51].

The correlation of phloem K and phloem sugar levels is still unclear and might differ between plants with symplasmic or apoplasmic phloem-loading strategies. Several studies indicate that a reduction of phloem K results in a concomitant decrease of transported assimilates. Mengel and Haeder [52] report that an increase of K from 0.4 mM to 1 mM in hydroponic nutrient solution increased the phloem sap exudation rate and sap sugar concentration of castor bean (*Ricinus communis* L.). In bean (*Vicia faba* L.), continued K-deficiency resulted in decreased sugar in phloem exudates and higher leaf sugar contents, suggesting impaired phloem loading and backlogging of sugars in the leaf mesophyll or the apoplasm, under low K [14]. Deeken et al. [43] demonstrated that phloem sap of Arabidopsis *akt2/3* mutants impaired in phloem K-loading transported lower amounts of both, K and sucrose in the phloem, as determined by aphid stylectomy. Comparing *P. major* whole leaf accumulation (Figure 2 and Figure 4) and phloem exudate data (Figure 5) indicated that homeostasis of K, sorbitol, and sucrose levels was much tighter regulated in the phloem than in the whole leaf. It is therefore highly likely that the different K-dependent accumulation of these substances occurred preferentially in the mesophyll cells of the leaf and steered responses related to stomata opening, photosynthesis, and vacuolar osmolyte accumulation. The homeostatic regulation of K in the phloem might be due to the adjusted regulation and activity of phloem K channels from the AKT family [16,43].

Cold stress impacts both assimilate flow and water uptake. This is because enzymatic reactions establishing root pressure and phloem sugar loading slow down at low temperatures and cannot keep up with water loss caused by transpiration. K is also a crucial factor regulating stomatal closure during water limiting conditions [29,53,54]. Growing *P. major* plants under different K soil substrate and transferring them to cold conditions (4 °C), we observed cold-induced wilting and water loss of leaves almost in all K conditions, but the severity of the water loss was dependent on the K soil level (Figure 6). The K availability influenced cold-dependent monosaccharide and sorbitol accumulation, and plants supplied with more K in the soil substrate were able to acclimate faster to cold temperatures (Figure 6A). The observed fast accumulation of maltose is most likely caused by rapid cold-induced downregulation of the plastid envelope located MALTOSE EXPORTER (MEX), and might therefore be restricted to chloroplast stroma [55,56]. However, maltose accumulation was not affected by the different K soil conditions. Interestingly, the starch levels steadily decreased over the whole time-period analyzed (Figure 6H). This was in contrast to Arabidopsis, where the starch levels maintained diurnal cycles of biosynthesis and breakdown in the cold [57]. The starch breakdown in the cold observed here rather resembled starch kinetics in cold-sensitive plants like sugar beet [41].

Interestingly, expression of genes encoding sorbitol and sucrose phloem loaders PMT1/2 and SUC2 were not significantly regulated by the different K soil conditions, under normal ambient temperatures (Figure 7). However, the expression of PMT1 and PMT2 was highly induced under cold condition but this cold-induction was dampened with increasing K level in the soil. Sorbitol transporters might be induced at low temperatures to enhance the transport capacity of the sugar alcohol required for the provision of sink organs and young leaves, with increased sorbitol for cold and freezing protection. The decrease in the expression levels of the SDH gene, encoding sorbitol-degrading sorbitol dehydrogenase, supports a scenario where the available sorbitol resources for sink supply increased during cold conditions. Increasing K levels in the mesophyll and presumably in the apoplast, established a raise in the electrochemical gradient Δψ between the apoplasm and companion cells. A steeper Δψ gradient possibly facilitates phloem sorbitol loading without the necessity to enhance the transport capacity for sorbitol, in the form of additionally deployed PMT transporters. Through such means, additional K supply could result in a more effective usage of the phloem-loading machinery and a faster response to stress conditions. These results indicated that the higher K supply enabled plants to respond faster to cold stress and suggest that the higher accumulation of sorbitol had a priming effect on the prevention of cold-stress-induced water loss.

## 4. Materials and Methods

### 4.1. Plant Material and Plant Growth

*P. major* plants used in this study were of an inbred line used in prior studies [11,26]. Plants were grown in growth chambers under short-day conditions (10 h light /14 h dark) at 120 µE and 21 °C. Vascular bundles and mesophyll tissue without vascular bundles were obtained from petioles, as published [58,59].

### 4.2. Soil Substrate Preparation

Plants were grown in pots (7.5 cm diameter) filled with a peat substrate, with an extremely low nutritional value (Zero-Soil, Hawita, Vechta, Germany). This substrate contained background levels of K lower than 0.1 g·kg^-1^ [33]. Three different soil substrate mixtures (K1 to K3) were prepared. The following stock solutions were used: NH_4_NO_3_ (60 g·L^−1^), KH_2_PO_4_ (40 g·L^−1^), K_2_SO_4_ (6 g·L^−1^), MgSO_4_ (20 g·L^−1^), H_3_BO_3_ (1.4 g·L^−1^), FeNa-EDTA (0.7 g·L^−1^), CuSO_4_·5H_2_O (0.8 g·L^−1^), ZnSO_4_·7H_2_O (0.9 g·L^−1^), MnCl_2_·4H_2_O (1.8 g·L^−1^), Na_2_MoO_4_ (7.5 mg·L^−1^). Per 20 kg unfertilized soil substrate, 200 mL of each of the stock solutions were added. A total of 1 L suspensions of each CaO (60 g·L^−1^) and CaCO_3_ (80 g·L^−1^) were directly mixed into 20 kg soil substrate. The soil substrate per nutrient mixtures were homogenized by repeated shoveling and dispersing of the mixtures. For the K-deficient K1 substrate, KH_2_PO_4_ was replaced with NaH_2_PO_4_ (40 g·L^−1^). For the K over-fertilization K3 substrate, KCl (70 g·L^−1^) was applied. The final calculated elemental concentrations in the soil substrates were as following: N (200 mg·kg^−1^), S (55 mg·kg^−1^), Mg (40 mg·kg^−1^), Ca (3978 mg·kg^−1^), P (93 mg·kg^−1^), K (27 mg·kg^−1^ for K1, 142 mg·kg^−1^ for K2; 500 mg·kg^−1^ for K3), B (2.5 mg·kg^−1^) Fe (0.92 mg·kg^−1^), Cu (2 mg·kg^−1^), Zn (2 mg·kg^−1^), Mn (5.2 mg·kg^−1^), and Mo (0.03 mg·kg^−1^).

### 4.3. Cold Acclimation Treatment

Plants were initially grown for 8 weeks in growth chambers, under short-day conditions (10 h light/14 h dark), at 110 µE (tubular fluorescent lamp light) and 21 °C. Plants were then transferred to a neighboring growth chamber with identical light conditions, and a temperature of 4 °C. Immediately prior to transfer and at one, two, and four days after transfer, whole shoots of six plants from each K soil substrate condition were harvested and the FW, DW, and water content were determined. Plants were harvested in the cold and the shoots were enclosed in pre-weighted 25 mL scintillation flasks. FW was determined and the plants were immediately stored at −80 °C. Plants were then freeze-dried for four days and used for DW and water-content determination. The freeze-dried material was ground in a ball mill and the powder used for metabolite and gene expression analysis.

### 4.4. PAM Measurements

Photosynthetic activity was measured using an Imaging-PAM *M-Series* system (Heinz Walz, Effeltrich, Germany), as described previously [11]. Measurements were performed with four to five plants per condition, and each plant was measured three times. After dark adaption of plants for 20 min, capacity of PSII was measured by saturation with a series of light pulses. Light curves were recorded by incrementally increasing light pulses with intensity from PAR (µmol photons m^−2^ s^−1^) 0 to PAR 726, in 14 steps. The dark interval between each light pulse was 20 s. Recorded fluorescence was used for calculation of the effective quantum yield of PSII, as described [11,41].

### 4.5. Gas Exchange Measurements

Plants were grown as described above for 6 weeks. Gas-exchange-related parameters were analyzed with a GFS-3000 system (Heinz Walz, Effeltrich, Germany). Measurements were performed with four to five plants per condition and each plant was measured three times (technical replicates). Individual plants were placed in a whole plant gas-exchange cuvette and the CO_2_-assimilation rate, respiration, leaf CO_2_ concentration, and stomatal conductance were recorded. Temperature, humidity, and CO_2_ concentrations of the cuvette were set to the conditions that the plants were grown in. Light respiration was measured at PAR 125 and dark respiration at PAR 0, over a time of 1 min for each plant. The CO_2_ concentration of the gas exchange chamber was set to 500 ppm, for both light and dark recordings. Each plant was measured three times with 30 s intervals between measurement, to allow the leaves to return to the stabilized value.

### 4.6. Phloem Exudates

Exudates were collected, as described in Pommerrenig et al. [26]. Exudates were collected for 10 h in 500 µL of 10 mM EDTA pH 8.0 solution. A total of 20 µL of a 1 to 3 dilution in water were used for sugar determination by ion chromatography. FWs of leaves were determined after exudation and used for calculation of obtained sugar amounts and exudation rate. Sugars in the exudates were determined using ion chromatography, as described below.

### 4.7. Soluble Sugar Extraction and Quantification

Freeze-dried plant material was ground to a fine powder using a ball mill (Retsch, Haan, Germany). About 15 mg of ground tissue were weighed into 2 mL screw-lid reaction tubes containing a steel ball and extracted with 900 µL of 80% ethanol p.a., on an overhead shaker, by inverting samples overnight at RT. Samples were centrifuged for 5 min at 11,000× *g* and 700 µL of the supernatants transferred to new 2 mL reaction tubes. Another 600 µL of 80% ethanol p.a. were added and the samples were extracted for 30 min at 50 °C, on a heating block, while shaking. The samples were again centrifuged as above and with 500 µL of the supernatant, combined with the already collected supernatants. Ethanol was removed in a vacuum concentrator (Eppendorf, Hamburg, Germany) and the pellets re-suspended in 500 µL ddH_2_O. Glucose, fructose, sucrose, sorbitol, and maltose were measured with ion chromatography, using an 871 Advanced Biosensor, 818 IC Pump, 837 De-gaser configuration from Metrohm (Herisau, Switzerland). A Metrosep Carb2 −250/4.0 column was used in combination with a Metrosep Carb2 Guard/4.0 guard column (both Metrohm, Herisau, Switzerland). A gold working electrode (2 mm diameter, Metrohm, Herisau, Switzerland) was used for amperometric detection. The eluent was 10 mM NaOAc and 100 mM NaOH. The flow rate was 0.6 mL min^−1^.

Freeze-dried plant material was ground to a fine powder using a ball mill (Retsch, Haan, Germany). About 15 mg of ground tissue were weighed into 2 mL screw-lid reaction tubes containing a steel ball and extracted with 900 µL of 80% ethanol p.a., on an overhead shaker, by inverting samples overnight at RT. Samples were centrifuged for 5 min at 11,000× *g* and 700 µL of the supernatants transferred to new 2 mL reaction tubes. Another 600 µL of 80% ethanol p.a. were added and the samples were extracted for 30 min at 50 °C, on a heating block, while shaking. The samples were again centrifuged as above and with 500 µL of the supernatant, combined with the already collected supernatants. Ethanol was removed in a vacuum concentrator (Eppendorf, Hamburg, Germany) and the pellets re-suspended in 500 µL ddH_2_O. Glucose, fructose, sucrose, sorbitol, and maltose were measured with ion chromatography, using an 871 Advanced Biosensor, 818 IC Pump, 837 De-gaser configuration from Metrohm (Herisau, Switzerland). A Metrosep Carb2 −250/4.0 column was used in combination with a Metrosep Carb2 Guard/4.0 guard column (both Metrohm, Herisau, Switzerland). A gold working electrode (2 mm diameter, Metrohm, Herisau, Switzerland) was used for amperometric detection. The eluent was 10 mM NaOAc and 100 mM NaOH. The flow rate was 0.6 mL min^−1^.

### 4.8. Chlorophyll and Pigment Analysis

Photosynthetic pigments were extracted from ground leave tissue, with 90% acetone/10% 0.2 M Tris/HCl pH 7.5, for 48 h, at 4 °C in the dark. Chlorophyll *a* and *b* were measured by the absorbance of the supernatant at 649 nm and 665 nm. The chlorophyll content was calculated using the formulas (Equations (1) and (2)):(1)$$\mathrm{Chl}\;a\;\left(\frac{\mu g}{\mathrm{mL}}\right)=11.63\times A665-2.39\times 649$$
and
(2)Chl b (μg/mL)=20.11×A649−5.18×A665

Violaxanthin, Antheraxanthin, and Zeaxanthin were analyzed through reverse-phase high-performance liquid chromatography, as described [48].

### 4.9. Statistical Analysis

Statistical analysis was performed by one-way ANOVA with post-hoc Tukey HSD test. A threshold of *p* = 0.05 was set for defining the significant differences. The testing was performed using the “One-Way ANOVA (ANalysis of VAriance) with post-hoc Tukey HSD (Honestly Significant Difference) Test Calculator for comparing multiple treatments” (https://astatsa.com/OneWay_Anova_with_TukeyHSD/). Boxplots were generated using BoxPlotR [60].

## 5. Conclusions

The present study provided evidence that *P*. *major* responded to K nutrition and performed better under low temperature, due to the regulation of carbohydrate metabolism, particularly sorbitol accumulation. Sorbitol might have a dual role in the cold acclimation process. First, sorbitol biosynthesis functions as electron sink via regeneration of NADPH/H^+^ to NADP^+^, during stress, alleviating high electron pressure from photosystems. Second, sorbitol, like other sugars might act as an ROS-quenching compound itself, due to the numerous hydroxyl groups and could be beneficial for absorption of free electrons during stress conditions. High K conditions promote such sorbitol accumulation. Our data link beneficial K nutritional effects for cold acclimation to sugar and sorbitol metabolism and offers a possible explanation of the physiological mode of action of K in abiotic stress response.

## Figures and Tables

**Figure 1 plants-09-01259-f001:**
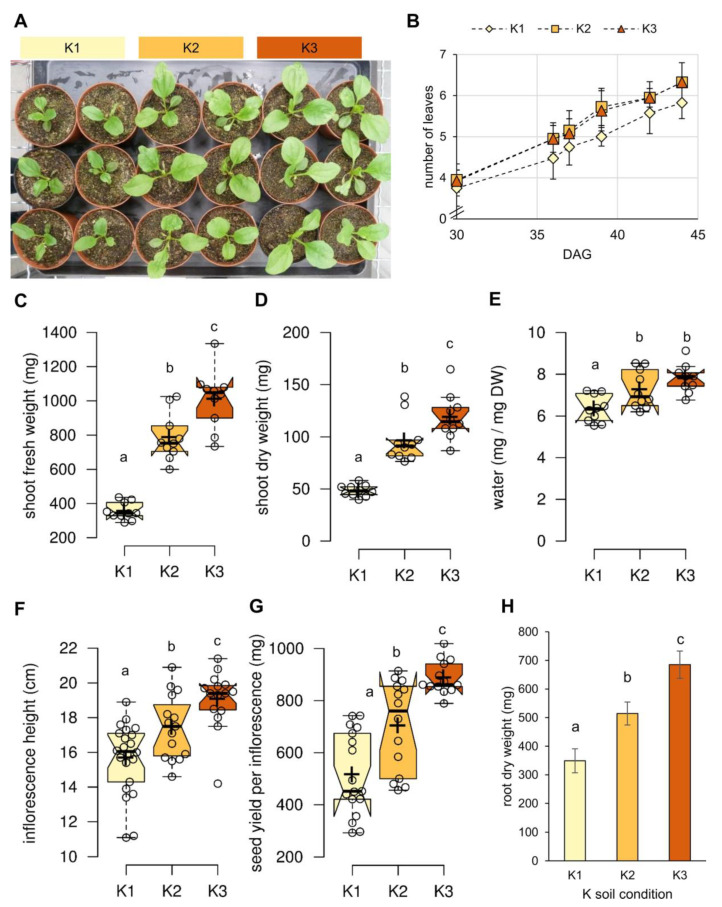
Effect of different soil K levels on biomass of *P. major*. (**A**) Exemplary pictures of 5-week old *P. major* plants grown under 3 different doses of soil K. (**B**) Development of leaf number over a time of 15 days (data represent means from *n* = 15 plants per data point ± SD). (**C**) Shoot fresh weight; (**D**) shoot dry weight; (**E**) shoot water content; (**F**) inflorescence height; (**G**) number of seeds per inflorescence, and (**H**) root dry weights of K1 to K3-grown plants (bars represent means from *n* = 6 roots per K condition ± SE). (**C**–**G**) Boxplots based on values from at least *n* = 12 plants per soil K level. Center lines within boxes represent median, crosses are the mean values. The notches are defined as +/− 1.58 × IQR/sqrt (*n*) and represent the 95% confidence interval for each median. Different letters denote significant changes between K treatments, according to one-way ANOVA with post-hoc Tukey HSD test (*p* = 0.05).

**Figure 2 plants-09-01259-f002:**
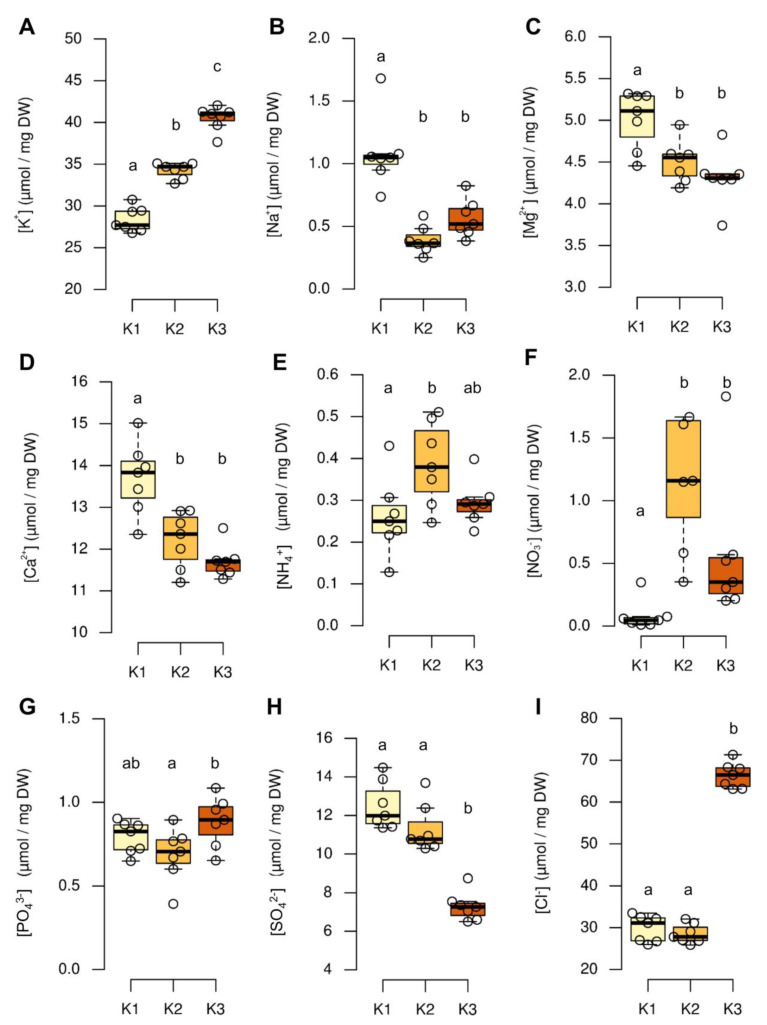
Effect of different soil K levels on nutrient concentrations in shoots of *P. major* plants. (**A**) Potassium content, (**B**) sodium content, (**C**) ammonium content, (**D**) magnesium content, (**E**) calcium content, (**F**) nitrate content, (**G**) phosphate content, (**H**) sulfate content, and (**I**) chloride content. Boxplots show distribution of *n* = 7 replicates. Center lines show the medians; box limits indicate the 25th and 75th percentiles. Different letters above boxes indicate significant differences between the K conditions, according to one-way ANOVA with post-hoc Tukey HSD test (*p* = 0.05).

**Figure 3 plants-09-01259-f003:**
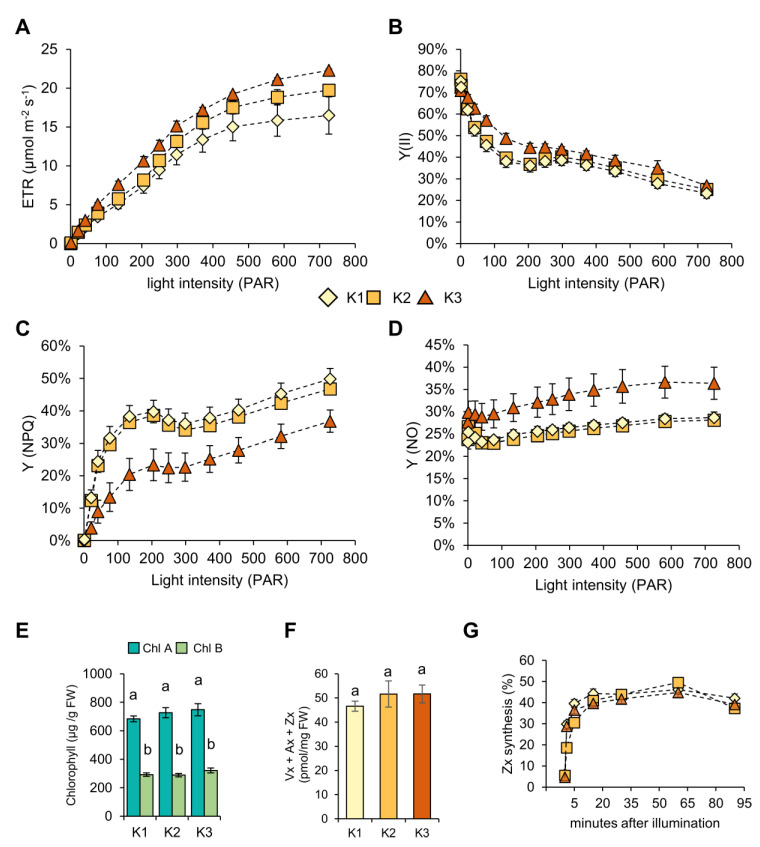
Effect of different soil K levels on photosynthetic parameters and photosynthetic pigments. (**A**) Electron transfer rate (ETR) describing how many electrons were generated from the collected photons at photosystem II, depending on the light intensity. (**B**) Quantum yield of photosynthesis [Y(II)], depending on the light intensity. (**C**) Non-photochemical quenching [Y(NPQ)], depending on the light intensity. (**D**) Non-regulated quenching [Y(NO)], depending on the light intensity. (**E**) Chlorophyll A and B contents of leaves. (**F**) Summarized content of violaxanthin (Vx), antheraxanthin (Ax), and zeaxanthin (Zx). (**G**) Zeaxanthin synthesis (= Zx/(Vx + Ax + Zx) × 100%) after transfer of leaves from dark to high light (750 µE). Four plants were analyzed per condition. Data points are means ± SE. Light curves were recorded using continuous light, incrementally increasing data in (**F**–**G**) are means from *n* = 5 replicates ± SE. Different letters above bars indicate significant differences, according to one-way ANOVA with post-hoc Tukey HSD test (*p* < 0.05).

**Figure 4 plants-09-01259-f004:**
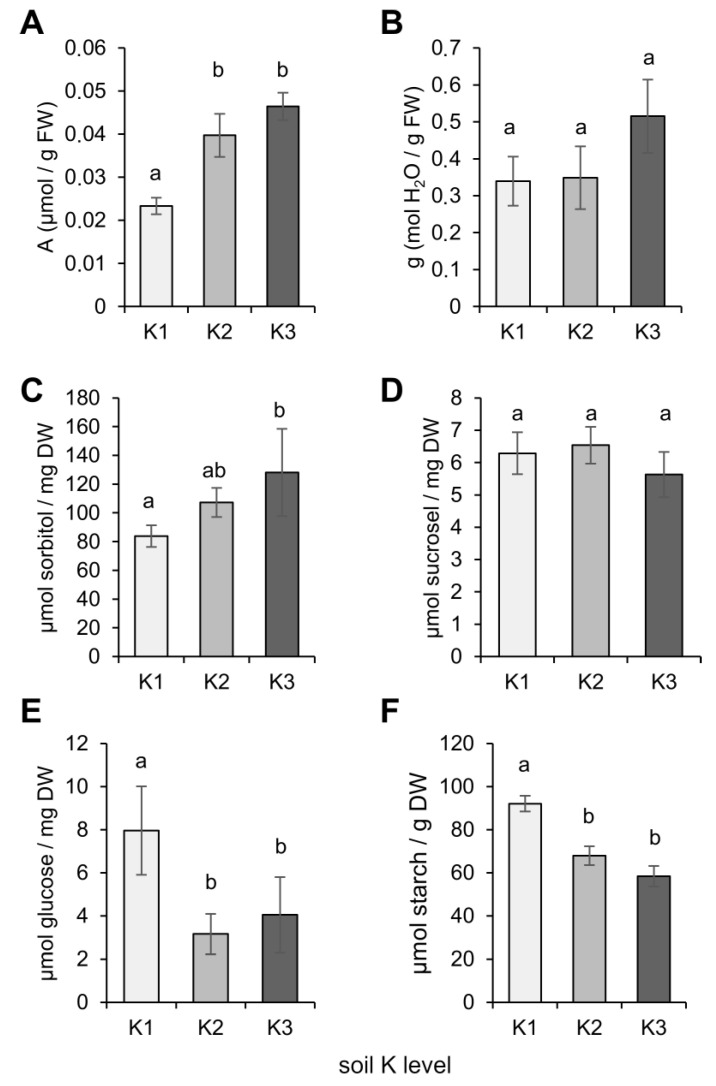
Effect of different soil K levels on the gas exchange parameters and the assimilation rate of *P. major* plants. (**A**) CO_2_ assimilation rate recorded in the light (*A*), (**B**) stomatal conductance (*g*) in the light, (**C**–**F**) assimilate contents of shoots—(**C**) sorbitol, (**D**) sucrose, (**E**) glucose, and (**F**) starch. Bars are means of *n* = 4 (**A**–**D**) or *n* = 5 (**D**–**F**) biological replicates ± SE. Different letters above bars indicate significant differences according to one-way ANOVA, with post-hoc Tukey HSD test (*p* < 0.05).

**Figure 5 plants-09-01259-f005:**
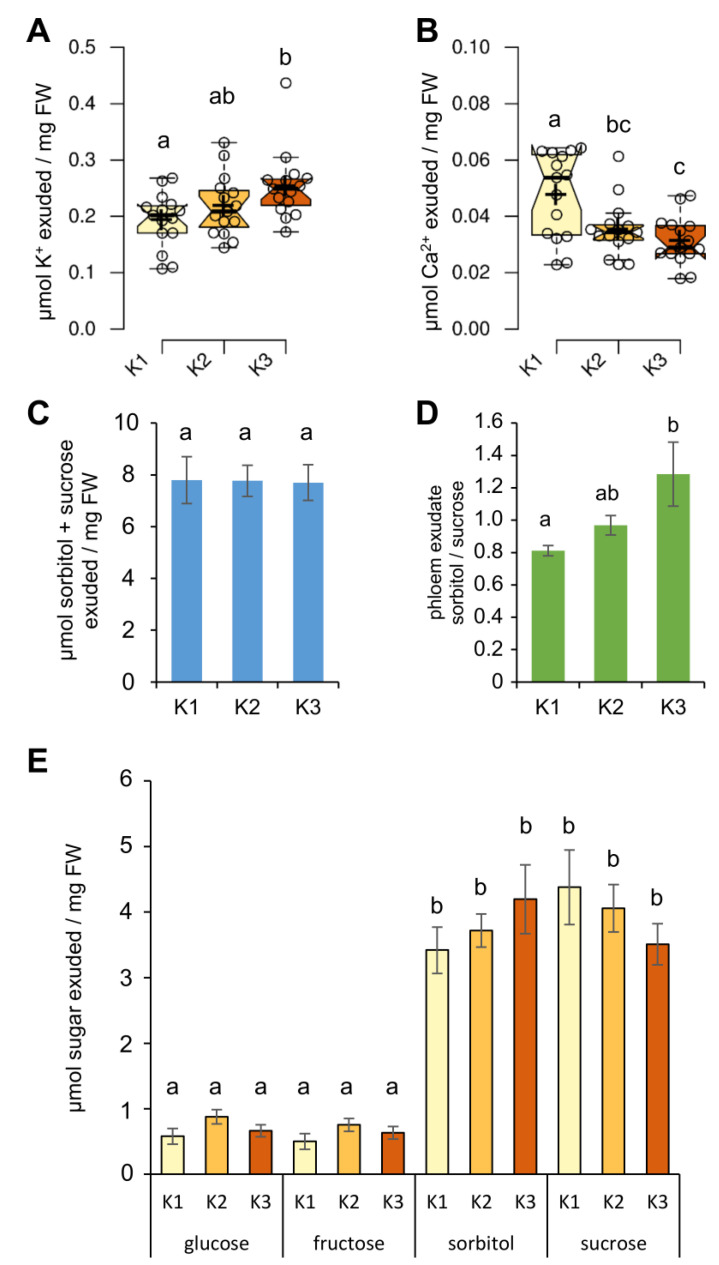
Effect of different soil K levels on phloem exudates. (**A**) K^+^ level; (**B**) Ca^2+^ level; (**C**) sum of sorbitol and sucrose; (**D**) sorbitol to sucrose ratio, and (**E**) sugar profile in phloem. Boxplots are based on values from *n* = 15 plants per soil K level. Center lines within boxes represent median, and crosses are mean values. The notches are defined as +/− 1.58 × IQR/sqrt (*n*) and represent the 95% confidence interval for each median. Bars are means from *n* = 15 replicates ± SD. Different letters indicate significant differences between K conditions, according to the one-way ANOVA with post-hoc Tukey HSD test (*p* < 0.05).

**Figure 6 plants-09-01259-f006:**
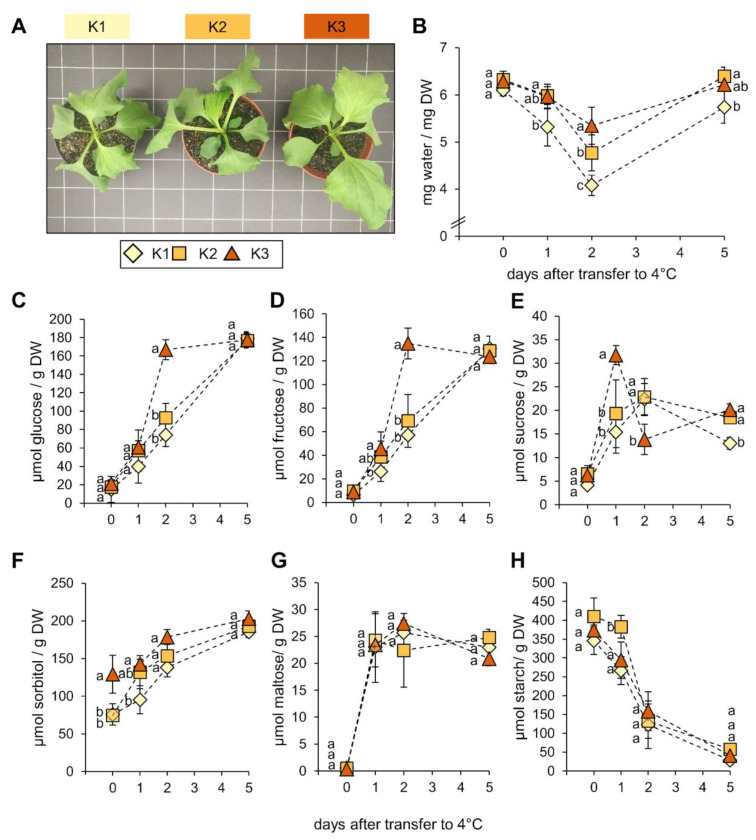
Effect of cold stress on plant growth and sugar accumulation in *P. major* plants grown on three different soil K levels. (**A**) Representative phenotypes of plants, two days after transferring from 20 °C to 4 °C. (**B**) Water content of shoots of plants before (0 days) and after transferring to cold condition. (**C**) Shoot glucose concentration; (**D**) shoot fructose concentration; (**E**) shoot sucrose concentration; (**F**) shoot sorbitol concentration; (**G**) shoot maltose concentration; and (**H**) shoot starch concentration during cold stress. Values are means from *n* = 5 plants ± SE. Different letters above bars denote significant differences according to one-way ANOVA with post-hoc Tukey HSD test (*p* < 0.05).

**Figure 7 plants-09-01259-f007:**
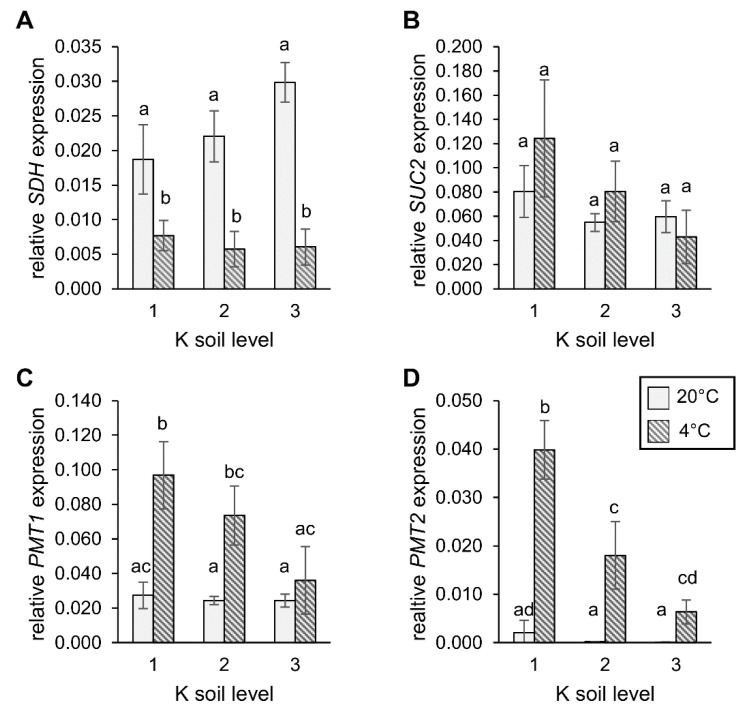
Effect of cold stress on the expression of sorbitol and sucrose-related genes. (**A**) Relative *SDH* mRNA levels, (**B**) relative *SUC2* mRNA levels, (**C**) relative *PMT1* mRNA levels, and (**D**) relative *PMT2* mRNA levels. Bars show means from *n* = 6 biological replicates ± SE. Different letters above bars denote significant differences between means, according to one-way ANOVA, with post-hoc Tukey HSD testing (*p* > 0.05).

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
