# Peer review of "Potassium Application Boosts Photosynthesis and Sorbitol Biosynthesis and Accelerates Cold Acclimation of Common Plantain (*Plantago major* L.)"

_plants, 2020, doi:10.3390/plants9101259_

Round 1

Reviewer 1 Report

The effect of potassium on carbohydrate metabolism and cold acclimation was investigated in several studies as stated by the authors in the first paragraph of the introduction. The comparison of three different potassium concentrations could give new information about its role in the studied physiological and biochemical processes. The authors should better emphasize the differences and similarities in the response of Plantago major to potassium in comparison with other plant species in which the effect of potassium was investigated.

Additional remarks:

  1. l. 127: Please, check the subtitle!
  2. 3a: The unit of ETR is not appropriate.
  3. l. 172: …depend on the epoxidation state…
  4. l. 236: extent
  5. l. 284: The differences, which are not significant, should not be mentioned in the text.
  6. l. 340-345, l. 378-385: These parts belong to the introduction.
  7. The grammatical and typing errors should be corrected.

Author Response

We thank the reviewer for the constructive comments and valuable corrections.

The reviewer also suggested to better compare the K effects observed for Plantago with other plant species. This comparison however cannot be easily done and is to our opinion a bit out of the scope of this work. Here we tried to address very specifically K effects in a polyol synthesizing plant and asked whether K-mediated faster acclimation to low temperatures could be linked to enhanced sorbitol biosynthesis. We describe how sorbitol biosynthesis functions as an electron sink during challenging cold stress conditions and that K promotes sorbitol biosynthesis. The only other polyol-synthesizing plant, where K-dependent growth and sugar accumulation has been – to our knowledge - investigated is pear (Pyrus bretschneideri) [Potassium influences expression of key genes involved in sorbitol metabolism and its assimilation in pear leaf and fruit; DOI: 10.1007/s00344-018-9783-1]. We acknowledge this publication in the discussion section. However, cold stress (or any other stress) has not been investigated in polyol synthesizing plants in a K-dependent manner. We find it therefore difficult to make rather general comparisons with other plant species. We believe that such endeavour would extend the already quite long discussion section. However, we went into detailed comparison, when individual questions became more specific, e.g. when we discussed K effects on phloem sugar transport. Here, we have mentioned very specifically studies which analysed transport of K and sugars though the phloem in the discussion section and compared our results there (see L. 356 to 371).

For the revised version of the manuscript we restructured significant parts of the introduction and discussion aiming to make our intention with the present study more clear.

 Additional remarks:

  1. l. 127: Please, check the subtitle!

Response: We changed the subheading to “K alters inorganic ion levels of leaves”

  1. 3a: The unit of ETR is not appropriate.

Response: The Reviewer is right. We changed the unit of the ETR to µmol m-2 s-1.

  1. l. 172: …depend on the epoxidation state…

Response: We changed the phrase accordingly.

  1. l. 236: extent

Response: We corrected the mistake

  1. l. 284: The differences, which are not significant, should not be mentioned in the text.

Response: The reviewer is right. We adjusted the paragraph and rephrased like follows:

“Air temperature did not significantly alter SUC2 expression levels, but increased expression of both sorbitol transporter genes (Figure 7B). Interestingly, this increase was dependent on the K soil level and higher K levels dampened the induction in the cold. (Figure 7C and 7D). These data showed that the K level changed the expression level sorbitol-related genes in a temperature dependent manner.”

  1. l. 340-345, l. 378-385: These parts belong to the introduction.

Response: We restructured the introduction and discussion sections. We believe that the revised versions of both sections meet suggestions from both Reviewer 1 and Reviewer 3.

  1. The grammatical and typing errors should be corrected.

Response: We went through the manuscript eliminating errors where we identified them.

Reviewer 2 Report

The manuscript presents interesting data, but need to be improved, English should be revised by a mother-tongue.

Please find below my suggestions:

L20: please change .."involvement of K in above processes.." with "involvement of K in the above processes";

L20: please change "We therefore studied K effects.." with "We, therefore, ..";

L128: please change "The different K supply resulted in regulation of.." with "The different K supply resulted in the regulation of..";

L166: please correct "of the photosynthesis machinary.." with " of the photosynthesis machinery..";

L175: please correct "The mesurements.." with "The measurements";

L224: I suggest to add letters also in figure 5E;

L264: please correct "and reached lowest " with "and reached the lowest ";

L309: please change "There are numerous studies indicating" with "Numerous studies are indicating".

Author Response

We thank the reviewer for his comments and corrections. We tried to implement as much as possible. We also went carefully through the manuscript eliminating typographic and grammar errors

L20: please change .."involvement of K in above processes.." with "involvement of K in the above processes";

Response: We changed the sentence accordingly.

L20: please change "We therefore studied K effects.." with "We, therefore, ..";

Response: We corrected the punctuation accordingly.

L128: please change "The different K supply resulted in regulation of.." with "The different K supply resulted in the regulation of..";

Response: We changed the sentence accordingly.

L166: please correct "of the photosynthesis machinary.." with " of the photosynthesis machinery..";

Response: We corrected the mistake.

L175: please correct "The mesurements.." with "The measurements";

Response: We corrected the mistake.

L224: I suggest to add letters also in figure 5E;

Response: We performed one-way ANOVA with post hoc Tukey test and added letters indicating significant differences to this subfigure

L264: please correct "and reached lowest " with "and reached the lowest ";

Response: We added the article accordingly.

L309: please change "There are numerous studies indicating" with "Numerous studies are indicating".

Response: We rephrased accordingly.

Reviewer 3 Report

This manuscript describes physiological responses to K in Plantago major as it relates to sorbitol biobiosynthesis and cold acclimation.  The research was conducted and analyzed very carefully. The manuscript perfectly readable, and good overall, but the subject matter is a bit difficult to follow at times. I have a few suggestions and comments.

The authors suggest that P. major is an interesting medicinal plant and useful model for plants containing high amounts of polyol sorbitols. I feel like these arguments could be strengthened in the Introduction and addressed more carefully in the Discussion. What does high amounts mean? Twice as much as other plants? Ten times as much as other plants? What other plants accumulate polyol sorbitol? 

I felt like the discussion was largely off the target subject (sorbitol biobiosynthesis and cold acclimation). The words "cold" and "sorbitol" were mentioned in the first brief summary paragraph of the Discuasion (Lines 399-308). The paragraphs from lines 309 to 345 do not mention cold or sorbitol, and we do not see the word "cold" mentioned again until line 378. I also thought that the discussion should discuss the results of the research as it relates to cold and sorbitol as reported in other studies.

The Conclusions have only two sentences. I agree with first sentence of Conclusions. However, arguments supporting the second sentence of the Conclusions were not adequately developed in the Discussion. The scope of the Conclusions is not balanced by the scope of the Discussion.

Author Response

Response: We thank the reviewer for the positive assessment and the constructive comments and suggestions. We tried to implement as much of these as possible.

The reviewer is right, that the discussion lacked a red arrow and some consistency. Having also specific comments of Reviewer 1 and the handling editor in mind, we restructured parts of the introduction and of the discussion. We also adjusted the scope of the conclusion.

For the revised version we tried to focus more on the cold treatment effects introducing cold stress earlier in the introduction. We make now more clear that cold stress induces a general problem for plants: Overflow of electrons at photosystems and therefore accumulation of reducing equivalents due to slow-down of Calvin-Benson cycle enzymes in the cold. This overflow of electron is also causative for the production of reactive oxygen species. We now describe in the discussion how our data indicate that sorbitol biosynthesis can function as an electron sink during such challenging conditions and therefore as a stress mitigation compound. We discuss that K promotes sorbitol and that observed alleviation of stress by increased K might be linked to increased sorbitol and sugar production.

The changes we integrated can all be tracked in revised version of the manuscript.